# Managing Severe Adverse Reactions to Biologicals in Severe Asthma

**DOI:** 10.3390/biomedicines11123108

**Published:** 2023-11-21

**Authors:** Radu-Gheorghe Balan, Diana Mihaela Deleanu, Irena Pintea, Carmen Teodora Dobrican Baruta, Milena Adina Man, Ioana Corina Bocsan, Ioana Adriana Muntean

**Affiliations:** 1Department of Allergology and Immunology, “Iuliu Hațieganu” University of Medicine and Pharmacy, Str. Croitorilor 19–21, 400058 Cluj-Napoca, Romania; balan.radu.gheorghe@elearn.umfcluj.ro (R.-G.B.); diana.deleanu@umfcluj.ro (D.M.D.); bocsan.corina@umfcluj.ro (I.C.B.); adriana.muntean@umfcluj.ro (I.A.M.); 2Department of Science and Technology, George Emil Palade University of Medicine, Pharmacy, Science and Technology, 540067 Târgu Mureş, Romania; 3Department of Allergology, “Professor Doctor Octavian Fodor” Regional Institute of Gastroenterology and Hepatology, 400162 Cluj-Napoca, Romania; 4Department of Internal Medicine, “Professor Doctor Octavian Fodor” Regional Institute of Gastroenterology and Hepatology, 400162 Cluj-Napoca, Romania; 5Department of Medical Sciences, “Iuliu Hațieganu” University of Medicine and Pharmacy, 400337 Cluj-Napoca, Romania; manmilena50@yahoo.com; 6Department of Pneumology, “Leon Daniello” Clinical Hospital of Pulmonology, 400332 Cluj-Napoca, Romania; 7Department of Pharmacology, Toxicology and Clinical Pharmacology, “Iuliu Hațieganu” University of Medicine and Pharmacy, 400337 Cluj-Napoca, Romania; 8Almedo—Allergology and Clinical Immunology Outpatient Clinic, 400394 Cluj-Napoca, Romania

**Keywords:** asthma, monoclonal antibodies, biologic therapy, hypersensitivity, desensitization

## Abstract

Background: The use of biological agents in the treatment of various inflammatory and malignancy conditions has expanded rapidly. However, these agents can induce hypersensitivity reactions, posing significant clinical challenges. Methods: We conducted a retrospective study that included nine patients with severe asthma who experienced hypersensitivity reactions to biological agents (omalizumab, benralizumab and dupilumab). Results: Hypersensitivity reactions to biologicals in severe asthma were observed in 9 of 68 patients treated. In five cases, treatment was stopped or changed to another available biological, and for four patients administered under close surveillance, titrated provocation or desensitization was applied. Successful desensitization was achieved in three of the patients, allowing them to continue therapy without adverse reactions. Improvements in asthma control were observed post-desensitization, leading to the reduced need for systemic steroid treatments and an increase in quality of life. Conclusions: This study highlights the importance of recognizing hypersensitivity reactions to biologicals to have an appropriate approach for patients with severe asthma. As an effective approach for patients experiencing hypersensitivity reactions to biological agents, desensitization allows treatment continuation.

## 1. Introduction

In recent years, there has been a rapid increase in the number of biological agents approved by pharmaceutical agencies across the globe for treating various inflammatory and malignancy conditions. As the use of these agents continues to expand, despite their therapeutic potential, their mechanism of action exposes them to the risk of immune-mediated effects, which can have significant implications [1,2].

To gain a deeper understanding of adverse reactions related to biological agents, it is essential to consider key differences between them and conventional drugs. Unlike most drugs, which are generally small molecules with molecular weights below 1 kDa (kilodaltons), biological agents are large-sized proteins designed to structurally resemble autologous proteins and have much larger molecular weights exceeding 1 kDa. Furthermore, drugs are chemically synthesized, while biological agents are produced using molecular genetic techniques and purified from genetically modified cells. Most biological agents require parenteral administration to avoid gastrointestinal degradation, whereas drugs can often be administered orally or parenterally and undergo metabolism. While drug metabolism can sometimes generate immunogenic intermediaries, biological agents are subject to processing but not metabolism. Ultimately, biological agents exhibit intrinsic immune-mediated effects because they originate from non-self proteins, a characteristic usually associated with smaller synthetic compounds in drugs [1].

Due to these differences, efforts have been made to categorize adverse reactions to biological agents using classification systems that emphasize their immune-target effects. In 2006, Pichler proposed such a classification scheme, which was later elaborated by Haussman and his colleagues [1,2,3]. This classification assigns Greek letters (alpha, beta, gamma, delta and epsilon) to various types of reactions (Table 1).

Beta-type reactions to biological agents involve hypersensitivity reactions, which can manifest as immediate or delayed responses. Factors that influenced these reactions include the type of immune response triggered: IgE (immunoglobulin E) mediated complement activation or immune complexes disease. Also, in the immune response towards biologicals, the degree of humanization of the monoclonal antibody and the presence of adjuvants or excipients should also be considered in the assessment of the reaction. Immediate reactions may involve the production of IgE antibodies targeting non-self peptide sequences, although immediate IgE-mediated hypersensitivity reactions are not commonly the primary cause of these reactions. Many patients can tolerate the same agent when administered at a slower rate or with premedication, such as antihistamines and steroids. However, IgE-mediated anaphylactic reactions have been reported with several biological agents, including infliximab, omalizumab and cetuximab, which were available early on the market, but also in new agents such as benralizumab and checkpoint inhibitors (nivolumab and ipilimumab). Studies have shown that IgE-mediated anaphylactic reactions to cetuximab are associated with pre-existing IgE antibodies against galactose-α-1,3-galactose. Common acute infusion reactions, largely comprising predictable and mild symptoms, represent most reactions to monoclonal antibodies. The underlying mechanism is not fully understood but may involve the release of proinflammatory cytokines [4,5,6,7,8,9].

Complement activation is also considered to play a role in immediate hypersensitivity reactions, as C3a (complement component 3 activated) and C5a (complement component 5 activated) cleavage products can directly stimulate mast cells and trigger mast cell activation independent of IgE. The degree of humanization of monoclonal antibodies has evolved, with fully humanized and human monoclonal antibodies having reduced immunogenicity compared to antibodies derived from mice. However, even humanized monoclonal antibodies contain non-self peptide sequences that can lead to the formation of anti-human antibodies. The consequences of these antibodies are usually delayed and involve the production of IgG antibodies, which can lead to drug inactivation but not necessarily severe symptoms. Complement is also believed to play a role in delayed reactions through the formation of immune complexes and serum sickness-like reactions. Some case reports have suggested T-cell-mediated hypersensitivity causing delayed maculopapular exanthems [3,4,5].

In severe asthma, there are different biological therapies available, as detailed in Table 2, such as anti-immunoglobulin E (anti-IgE) therapy (omalizumab), anti-interleukin-5 (anti-IL-5) (mepolizumab, reslizumab), anti-alpha subunit-IL-5 receptor (benralizumab), an anti-alpha subunit of the IL-4 receptor/IL-13 receptor (dupilumab) or anti-thymic stromal lymphopoietin (TSLP) (tezepelumab). The recommendation for a specific biological action depends on several factors such as their age, serum total IgE level, eosinophils level and the presence of type-2 asthma phenotype. The selection of treatment is also performed according to national protocols and availability of them in each country [10,11].

It is important to note that hypersensitivity reactions, including IgE-mediated ones, can present symptoms that overlap with common infusion reactions (such as gastrointestinal symptoms, dyspnea, transient skin redness, back pain) and symptoms of the underlying disease. Symptoms suggestive of hypersensitivity reactions may include urticaria with/without angioedema, wheezing, frequent cough, severe rash or anaphylactic symptoms. Hypersensitivity reactions to biological agents appear to be less common and have been reported for various agents, including omalizumab, natalizumab, rituximab, dupilumab and cetuximab [4,12,13,14,15,16,17,18].

Desensitization, as a management strategy for hypersensitivity reactions to biological agents, has emerged as an option, mainly when other biologicals are not available, but has yielded varying results [3,4].

In this study, we evaluated the hypersensitivity reactions to biological recommended for severe asthma that were available in our country in 2023; reactions were evaluated in three clinical settings specialized in severe asthma treatment.

## 2. Materials and Methods

We present a case series of adverse reactions to monoclonal antibodies (MoAb) used in severe asthma. From a group of 68 patients with severe asthma (treated in the Pneumology Department, the Allergology Department, Almedo Clinic), we evaluated 9 patients with adverse reactions requiring hospitalization or emergency room visits.

Out of the nine patients, three underwent successful desensitization to biologicals used in the treatment of severe asthma: two to omalizumab and another to dupilumab. For data collection, we reviewed the electronic and physical medical records.

Our study received approval from the Institutional Review Board at our facility (no. 10577/09.12.2021), and we obtained informed consent from all patients before proceeding with any investigation. The desensitization procedures took place between 2019 and 2022 at the Allergology Department of IRGH Cluj-Napoca, a specialized unit for treating allergic drug reactions.

## 3. Results

Of 68 patients with severe asthma under biologicals (from 2016 to 2023 in all three centers), 14 patients (20.5%) are treated with omalizumab, 16 patients with dupilumab (23.5%) and 38 patients (56%) with benralizumab. Hypersensitivity reactions are described below for each patient and summarized in Table 3.

### 3.1. Patient 1

A 60-year-old female with a history of severe persistent asthma since childhood had a moderate obstructive pattern in baseline spirometry with significant reversibility (FEV1 (Forced Expiratory Volume in the 1st second) = 57%, with an 18% bronchodilator response). After experiencing multiple ICU admissions and exacerbations necessitating oral steroids, she was started dupilumab 200 mg, resulting in significant improvement in asthma control after the first month of treatment. However, 2 h after the third dose of dupilumab, she developed a systemic hypersensitivity reaction to dupilumab. This reaction included facial edema and paresthesia with an extended local reaction at the site of administration in the absence of significant lung function impairment (FEV1 = 55%). While under the care of the Leon Daniello Pneumology Hospital, she was treated with hydrocortisone, salmeterol and diphenhydramine, with slow remission of symptoms during the following three days. Dupilumab was stopped, and OCS (oral corticosteroids) were added.

Two months later, she was referred to our center for testing and possible desensitization to dupilumab. Skin prick testing with undiluted dupilumab (200 mg/mL) yielded a weak irritation reaction (wheal of 2–3 mm in the absence of erythema or pruritus), and intradermal testing with concentrations starting from 0.2 mg/mL to 20 mg/mL (1:1000, 1:100, 1:10) were all negative (Table 3). Desensitizing was attempted with progressive administration of dupilumab diluted in saline solution every 30 min, according to Table 4. However, after receiving the fifth dose (40 mg, total dose of 70 mg), she developed undereye edema; desensitization was paused, and the patient was monitored for further developments. The next day, we continued desensitization according to Table 4 up to a cumulated dose of 200 mg dupilumab, which she tolerated. The protocol was completed without further complications, and she has since continued receiving 200 mg dupilumab every two weeks without any hypersensitivity reactions. She was able to taper off systemic steroids, and her asthma was well controlled.

### 3.2. Patient 2

A 31-year-old female had severe, persistent asthma and a history of two episodes of severe anaphylaxis to tree nuts, with baseline spirometry revealing a moderate obstructive pattern with significant reversibility (FEV1 = 63%, with a 21% bronchodilator response). She exhibited sensitivity to house dust mite allergens, various pollens and molds with a total IgE level of 487 IU/mL. Her asthma required daily prednisone in addition to high-dose inhaled corticosteroids combined with long-acting beta-agonists for control. After starting omalizumab, she was able to discontinue daily oral corticosteroids. However, after 3 months of omalizumab therapy, she presented an episode of grade II anaphylaxis [19]: generalized urticaria, facial edema, shortness of breath and extended local reaction at the site of administration immediately after injection.

She was referred to our center for testing and possible desensitization to omalizumab. Our target dose for desensitization was 300 mg out of her regular dose of 450 mg per month. Skin prick testing with undiluted omalizumab (150 mg/mL) was positive (Table 3). The desensitization was performed according to the protocol detailed in Table 5 over the course of two days. She completed the protocol without complications and has continued monthly treatments without further reactions. Her asthma has improved, allowing her to discontinue administration of oral corticosteroids (OCS).

### 3.3. Patient 3

A 40-year-old man with severe uncontrolled persistent asthma had baseline spirometry revealing mild restriction and moderate obstructive pattern without significant reversibility (FEV1 = 68%, with a 9% bronchodilator response). He also presented NSAID (non-steroidal anti-inflammatory drugs) exacerbated respiratory disease and chronic rhinosinusitis with nasal polyposis. He exhibited sensitivity to house dust mite allergens and grass pollen and had a history of grade II food-related anaphylaxis (peanuts, shrimp and pumpkin seeds). The total IgE level was 1434 IU/mL. His asthma required high-dose inhaled corticosteroids combined with long-acting beta-agonists and leukotriene receptor antagonists, and it was still uncontrolled. After the first dose of omalizumab, within 20 min, he experienced dyspnea, wheezing, angioedema and conjunctivitis; omalizumab was stopped.

He was referred to our center for testing and possible desensitization to omalizumab. Our target dose was 300 mg out of his regular dose of 450 mg per month. Skin prick testing with undiluted omalizumab (150 mg/mL) and intradermal testing with diluted omalizumab in saline (15 mg/mL, 1:10) were all negative (Table 3). Since skin prick tests were negative, we began the titrated provocation protocol detailed in Table 6. He completed the protocol without complications, and he has continued monthly treatments with the indicated dose without further reactions. His asthma has improved, allowing him to discontinue administration of oral corticosteroids (OCS).

### 3.4. Patient 4

A 53-year-old man with severe, persistent asthma: baseline spirometry revealed mild restriction and moderate obstructive pattern without significant reversibility (FEV1 = 61%, with a 7.5% bronchodilator response) and chronic rhinosinusitis with nasal polyposis. His asthma required high-dose inhaled corticosteroids combined with long-acting beta-agonists for control. Treatment was initiated with benralizumab 30 mg/month. One hour after the first dose, the patient experienced severe headache, hypertension (200/120 mmHg), tachycardia (120 bpm) and, later that evening, associated arthralgia and myalgia. Following emergency cardiological investigations, the patient started treatment with antihypertensive medication. Biological treatment with benralizumab was continued a month later without any further adverse effects.

### 3.5. Patient 5

A 48-year-old male with severe, persistent asthma with baseline spirometry revealing moderate–severe obstructive pattern with significant reversibility (FEV1 = 54%, with a 20% bronchodilator response) had presented grade III anaphylaxis to a bee sting and grade IV anaphylaxis to penicillin and ampicillin [19]. His asthma remains uncontrolled despite treatment with high-dose inhaled corticosteroids combined with long-acting beta-agonists and leukotriene receptor antagonists, as well as frequent courses of high-dose oral steroids. Treatment was initiated with omalizumab (300 mg q2wk). Five years of omalizumab treatment was tolerated. In the 5th year of treatment, 20 min after administration, the patient exhibited severe headache, vertigo, severe arterial hypotension and wheezing, interpreted as a grade III/IV anaphylactic reaction. Symptoms remitted after epinephrine administration, and he was hospitalized (for a total of 1000 mcg, epinephrine i.m.). Omalizumab was stopped, and OCS were added again for 6 months until dupilumab treatment was initiated when it was approved in Romania. The patient received dupilumab and tolerated the treatment with cessation of OCS.

### 3.6. Patient 6

A 30-year-old female with severe, persistent asthma presented baseline spirometry, revealing a moderate obstructive pattern with significant reversibility (FEV1 = 64%, with a 13% bronchodilator response) during asthma initial evaluation. She exhibited sensitivity to house dust mites and dog dander. Her asthma was still uncontrolled (FEV1 = 68%, ACT (asthma control test) = 10 maximum points) despite treatment with high-dose inhaled corticosteroids combined with long-acting beta-agonists and leukotriene receptor antagonists, as well as frequent courses of high-dose oral steroids. Treatment was initiated with omalizumab (300 mg q2wk). Three days after the first injection of omalizumab, the patient exhibited flu-like symptoms: myalgia, arthralgia and painful laterocervical adenopathy with generalized urticarial lesions, which needed a course of 4 weeks of OCS until remission. Subsequent attempts at continuing biological treatment with half of the dose led to the same outcome. The pattern of symptoms was interpreted as serum sickness due to omalizumab; thus, cessation of treatment was chosen.

### 3.7. Patient 7

A 50-year-old female with severe, persistent asthma and baseline spirometry revealing a moderate obstructive pattern with significant reversibility (FEV1 = 68%, with an 18% bronchodilator response) exhibited sensitivity to bees and yellow jacket venom. Her asthma required high-dose inhaled corticosteroids combined with long-acting beta-agonists and leukotriene receptor antagonists for control, as well as frequent courses of high-dose oral steroids. Treatment was initiated with omalizumab (300 mg q2wk). Two days after the second injection of omalizumab, the patient exhibited intense asthenia, nausea and dyspnea, which persisted for 3 weeks and made her unable to attend work; an OCS course for 2 weeks was helpful for her asthma and asthenia. The pattern of symptoms was interpreted as an adverse reaction to omalizumab. Other neurologic, rheumatologic or infectious diseases were excluded. No other biologic agent was initiated, despite the aggravation of her asthma, and several courses of OCS were needed.

### 3.8. Patient 8

A 47-year-old female with severe, persistent asthma and baseline spirometry revealing a moderate obstructive pattern with significant reversibility (FEV1 = 56%, with a 20% bronchodilator response) exhibited sensitivity to grass pollen. Her asthma required high-dose inhaled corticosteroids combined with long-acting beta-agonists and leukotriene receptor antagonists, as well as frequent courses of high-dose oral steroids, and still was uncontrolled. Treatment was initiated with benralizumab (30 mg q8wk). Five minutes after the eighth dose of benralizumab, the patient exhibited loss of consciousness (30 s), same-level fall, snoring and diaphoresis. After 24 h, she also develops mild urticaria. She was admitted to the ER, and the neurological assessment excluded an underlying neurological pathology. Symptoms remitted after systemic corticosteroids and supportive treatment.

Skin prick testing with dilute and undiluted benralizumab (30 mg/mL, 1:1; 3 mg/mL, 1:10) as well as intradermal testing with a concentration of dilute benralizumab in saline (3 mg/mL, 1:10), were performed, and all were negative (Table 3). Twenty-four hours after skin prick and intradermal tests, she developed generalized urticaria that ceased in twenty-four hours without treatment. Benralizumab was stopped.

### 3.9. Patient 9

A 68-year-old female with severe allergic asthma has had multiple allergen sensitizations since she was 20 years old (house dust mites, cat, grass pollen, birch pollen, cockroaches), with also two episodes of anaphylaxis in the last 5 years from nuts and banana. At 40 years old, baseline spirometry revealed a moderate obstructive pattern with significant reversibility (FEV1 = 62% from predicted, with a 14% bronchodilator response). Her asthma required high-dose inhaled corticosteroids combined with long-acting beta-agonists and leukotriene receptor antagonists, as well as frequent courses of high-dose oral steroids in the last year, and despite this treatment, asthma was still uncontrolled (FEV1= 67% from predicted, ACT test under 15 points 6 months before biological treatment initiation). Treatment was initiated with dupilumab (200 mg q2wk) with improvement in asthma control and lung function after the first doses and continuing 1 year after. After the 11th month of treatment, the patient developed large local reactions at the injection site (erythema, itch, edema around 5 mm). Forty-eight hours after the dose of dupilumab in the 15th month, the patient exhibited maculopapular rash (as shown in Figure 1), arthralgias, bloating, shivers and abdominal pain. She was presented in the ER; lab tests were performed for infectious disease screening, and inflammatory status showed no abnormalities. The dermatological evaluation was performed to exclude other dermatological diseases, but drug-induced maculopapular rash was confirmed. Symptoms remitted after systemic corticosteroids, antihistamines and supportive treatment. Dupilumab was stopped, and an allergy work-up was made after 4 weeks.

Skin prick testing with undiluted dupilumab after 4 weeks (200 mg/mL, 1:1) and intradermal testing with a concentration of dilute dupilumab in saline (20 mg/mL, 1:10) were performed and were all negative (Table 3). The patient tolerated several courses with vitamins from group B with Polysorbate 80 as an excipient. It is necessary to evaluate the risk/benefit ratio in order to purchase the titrated provocation; because severe asthma was controlled after dupilumab, we obtained informed consent and began the provocation test. This type of delayed hypersensitivity does not always present with a positive skin test, and provocation is the gold standard diagnosis method. The patient followed step 1 (day 1) from Table 3 at a cumulative dose of 20 mg dupilumab; after 2 h, the patient presented with shivers, lingual edema, metallic taste and a decrease in arterial tension (30 mmHg difference from baseline value); thus, the provocation test was stopped. Symptoms remitted after epinephrine, antihistamines, saline solution and systemic i.v. corticosteroids (8 mg dexamethasone).

## 4. Discussion

Biologicals in the treatment of severe asthma were introduced in Romania in 2016 (omalizumab), 2019 (benralizumab) and 2022 (dupilumab), with an increase in quality of life for those patients. This treatment is “life changing” for severe asthma patients, so the appearance of adverse reactions as hypersensitivity reactions could lead to important problems.

Although monoclonal antibodies have demonstrated their efficacy in clinical studies for severe asthma, several factors must be considered when initiating and monitoring biological treatment in severe asthma. In this clinical decision process, the risks/benefits of biologic therapy need to be understood to adequately counsel patients and appropriately monitor for potential adverse events. Also, because biologicals in severe asthma improve the level of control and quality of life, cessation should be considered as a last resort, especially when no other biological therapy is available, and desensitization protocols should be carefully implemented [20,21,22,23,24,25,26].

**Omalizumab** has been associated with an increased risk of anaphylaxis, and the product label includes a black box warning from the FDA for anaphylaxis, so an emergency kit with epinephrine is mandatory for users. The diagnosis of anaphylaxis is often difficult in severe asthma patients. Also, asthma increases the risk of severe anaphylaxis. On the other hand, omalizumab is useful in the treatment of severe IgE-mediated anaphylaxis, and it is used in desensitization protocols for other drugs or pre-treatment in allergen immunotherapy [20,21,22,23,24,25,26,27].

Risk factors for developing anaphylaxis to omalizumab include a prior history of anaphylaxis, which was present in the clinical history of our three patients with omalizumab anaphylaxis [28,29].

Diagnostic hypersensitivity reactions to monoclonal antibodies are difficult; testing for IgE-mediated hypersensitivity reactions may involve skin and intradermal tests, but it is crucial to determine non-irritating concentrations for drug testing. Omalizumab is the only biological agent for which non-irritating concentrations have been systematically established, but there are unmet needs to establish non-irritating concentrations for other biologicals. Various dilutions of omalizumab for skin and intradermal tests were studied in 2010 to ensure safety and interpretable results for immediate IgE-mediated hypersensitivity reactions. It was found that dilutions in sterile water caused irritant reactions, leading to the subsequent use of saline dilutions. An established non-irritant concentration for omalizumab testing with saline dilution is 1:100,000 (equivalent to a concentration of 1.25 mg/mL). However, the utility of omalizumab skin testing and additional data on positive and negative predictive values remain unknown [30,31].

Desensitization is a procedure performed only by specialized centers, where patients receive increasing doses of a drug to which they had previously experienced a hypersensitivity reaction until the target therapeutic dosage is achieved. The exact mechanism by which desensitization occurs is not entirely understood but involves temporary tolerance of mast cells and basophils to the drug involved in the hypersensitivity reaction. Successful desensitization protocols have been published for other monoclonal antibodies, especially for omalizumab, which has a high rate of anaphylaxis [29,30,31]. In patients with a negative skin test or minimal risk, we chose a titrated provocation, which leads to a rapid increase in the amount of drug up to the therapeutical dose.

Several case reports describe the onset of adverse events like arthralgia after use of omalizumab. Specifically, serum sickness-like reactions have been described, as we also reported in our communication [26,32,33].

**Benralizumab** is mainly indicated in eosinophilic severe asthma; hypersensitivity reactions (urticaria, polyangiitis, and even anaphylaxis in one case) were reported in placebo-controlled trials. In our patient in whom we suspected anaphylaxis to benralizumab, serum tryptase was not evaluated immediately after the reaction, which could be helpful, but it is not a common attitude in ER (emergency room) because it is not available. In this case, due to the increased severity of the reaction, benralizumab was ceased. Several case reports describe the development of inflammatory disorders in patients receiving benralizumab, including cytokine-release reaction, which we also described in one patient, but with the disappearance of the symptoms as we continued the treatment. Also, we described possible anaphylaxis to benralizumab; only another case was reported, so we considered that anaphylaxis is rare [26,29,34].

**Dupilumab** is newly introduced in our country, but hypersensitivity reactions occur in less than 1% of patients, including generalized urticaria, serum sickness, rash, erythema nodosum and anaphylaxis. Generally, dupilumab is well tolerated, but eosinophilia and conjunctivitis are reported more commonly [26,29,35]. Diagnosis of hypersensitivity reactions is difficult. Non-irritating concentrations for dupilumab skin tests have not yet been established, although some case series with a limited number of patients determined concentrations of undiluted dupilumab (150 mg/mL) for skin prick test and a dilution of 1:10 in saline for intradermal testing [32]. Since, for many patients, treatment continuation is particularly important for severe asthma management, in low-risk patients such as our case, we can safely readministered the drug.

Excipients are used as drug preservatives in many products, like polysorbate 80 from omalizumab and dupilumab, which may induce allergic reactions, which is a problem that needs to be discussed. Of course, patients with multiple hypersensitivity reactions to drugs containing polysorbate 80 are the candidates for this type of allergy work-up, but there is still no standard evaluation or commercial test substance for skin tests. Drug provocation with two different drugs containing polysorbate 80 should be considered in these patients, but it is time consuming and has a high cost and risks for patients [35,36].

In general, biologicals are well tolerated in patients with severe asthma, but real-life experiences are still published. New studies showed that there is an increased risk of adverse events related to the drugs for patients with biologicals in severe asthma [10,36,37]. Omalizumab presents an increased risk for anaphylaxis, as we observed among our patients. We need a good surveillance system in adverse events reporting that is voluntary and easy to manage, which is not accurately described in the literature.

We share our experience with successful desensitization or titrated provocation in three patients with severe, previously steroid-dependent asthma who experienced hypersensitivity reactions to biologicals. They have continued therapy for at least 12 months post-desensitization without experiencing adverse reactions. Following successful desensitization, all patients showed significant improvements in asthma control and were able to discontinue or reduce oral steroids.

One limitation of our study is that none of the patients had their serum tryptase levels checked after their reactions, which could have supplied additional evidence of mast cell degranulation in immediate-type hypersensitivity reactions. Moreover, we did not perform skin testing for excipients, for example, polysorbate 80, an excipient in both omalizumab and dupilumab, known to cause hypersensitivity reactions, as they are not commercially available for testing. There are studies that reveal these problems and the necessity to make registries for harmonized reports of hypersensitivity reactions to biological therapies [37,38]. Another limitation of our study is the small sample size since the treatments are still recently introduced in clinical practice, and severe asthma is observed in a small percentage of asthmatic patients [10].

## 5. Conclusions

In conclusion, it is important to report adverse events and follow up with patients with severe asthma under biological treatment to make a correct decision. Desensitization or titrated provocation emerges as a crucial therapeutic choice for managing patients who exhibit hypersensitivity reactions to biologicals because, in many cases, biologicals are “a life-changing” treatment inducing asthma remission. This approach can lead to substantial enhancements in asthma control and quality of life while reducing the necessity for systemic steroid treatments.

## Figures and Tables

**Figure 1 biomedicines-11-03108-f001:**
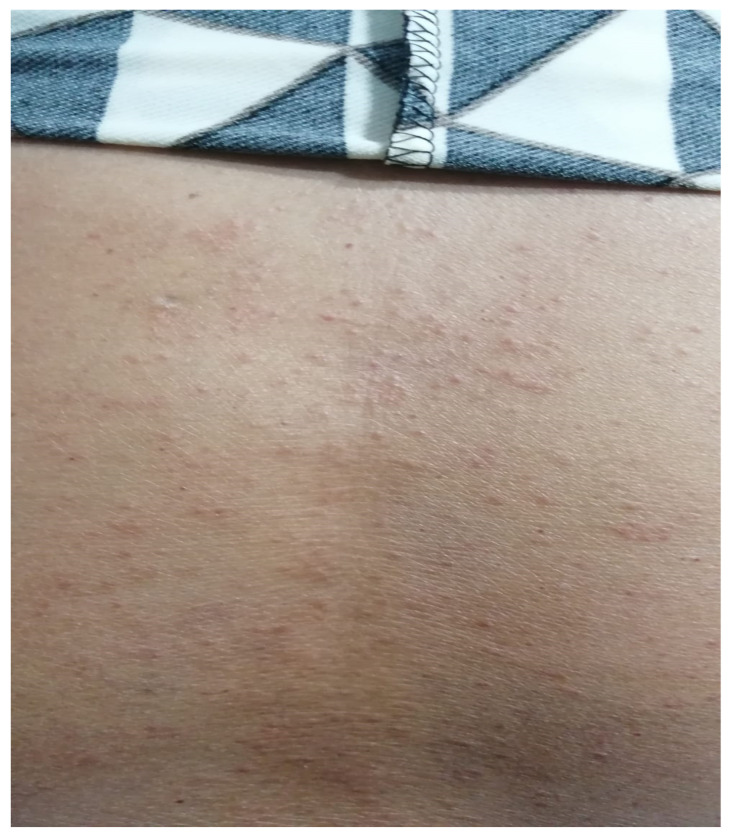
Skin aspect on the back, extended to all skin.

**Table 1 biomedicines-11-03108-t001:** Adverse reactions to biological agents [1,2].

Type	Example Reaction (Causative Drug)
α: Overstimulation	Cytokine release syndrome (cytokine storm) (muromunab, TGN1412)
β: Hypersensitivity	Common acute infusion reactions (rituximab), delayed infusion reactions (etanercept, adalimumab), anaphylaxis (muromunab, cetuximab, omalizumab)
γ: Cytokine of immune imbalance	
Immunodeficiency	Increased risk of tuberculosis—anti-TNF (tumor necrosis factor) agents Hypogammaglobulinemia (rituximab)
Autoimmunity	Systemic lupus erythematosus or vasculitis: IFN-γ(interferon gamma)
Atopic disorders	Atopic dermatitis (anti-TNF agents)
δ: Cross-reactivity	Acne from anti-EGFR (epidermal growth factor receptor): cetuximab, panitumumab
ε: Nonimmunological side effects	Neuropsychiatric side effects including confusion or depression (IFN-α)

**Table 2 biomedicines-11-03108-t002:** Biologicals available for severe asthma in Europe in 2023 (alphabetical order) [11].

Generic Name	Trade Name	Target
Benralizumab	Fasenra^®^	alpha subunit of the IL5R
Dupilumab	Dupixent^®^	alpha subunit of the IL-4R and the IL-13R
Mepolizumab	Nucala^®^	IL5
Omalizumab	Xolair^®^	IgE
Reslizumab	Cinqair^®^	IL5
Tezepelumab	Tezspire^®^	TSLP

**Table 3 biomedicines-11-03108-t003:** Summary of investigations in 9 cases of MoAb (monoclonal antibodies) adverse reactions.

Patient	Age (y)/Sex	Index Reaction to MoAb	Drug Cause	SPT	IDT	Follow-Up Attitude
1	60 F	AFX/UNK	Dupilumab	UD negative	1:1000 negative	Desensitization/itrated provocation with successful continuation of treatment
					1:100 negative	
					1:10 negative	
2	31 F	AFX	Omalizumab	UD positive	ND	Desensitization with successful continuation of treatment
3	40 M	AFX/UNK	Omalizumab	UD negative	1:10 negative	Titrated provocation with successful continuation of treatment
4	53 M	AR	Benralizumab	ND	ND	Successful continuation of treatment
5	48 M	AFX	Omalizumab	ND	ND	Treatment cessation
6	30 F	SS	Omalizumab	ND	ND	Treatment cessation
7	50 F	Severe adverse reaction	Omalizumab	ND	ND	Treatment cessation
8	47 F	AFX/UNK	Benralizumab	UD negative	1:10 negative	Treatment cessation
				1:10 negative		
9	68 F	Delayed hypersensitivity (maculopapular rash with arthralgias and abdominal pain)	Dupilumab	UD negative	1/10 negative	Treatment cessation

AFX, anaphylaxis; F, female; M, male; UD, undiluted; ND, not done; SS, serum sickness; UNK, unknown; AR, adverse reaction.

**Table 4 biomedicines-11-03108-t004:** Proposed dupilumab desensitization protocol/titrated provocation.

Day	Dose	Solution Concentration (mg/mL)	Dosage (mg)
1	1	2	0.2
	2	20	0.4
	3	200	10
	4	200	20
	5	200	40
		Cumulative dose (mg)	70.00
2	6	200	20
	7	200	40
	8	200	80
	9	200	40
		Cumulative dose (mg)	200

**Table 5 biomedicines-11-03108-t005:** Omalizumab desensitization protocol.

Day	Dose	Solution Concentration (mg/mL)	Dosage (mg)
1	1	1.5	0.15
	2	15	1.5
	3	150	4.5
	4	150	22.5
	5	150	37.5
		Cumulative dose (mg)	66.15
2	6	150	15
	7	150	22.5
	8	150	37.5
	9	150	75
		Cumulative dose (mg)	150

**Table 6 biomedicines-11-03108-t006:** Omalizumab titrated provocation (minimal risk patients, negative skin tests).

Day	Dose	Solution Concentration (mg/mL)	Dosage (mg)
1	1	150	15
	2	150	30
	3	150	45
	4	150	60
	5	150	150
		Cumulative dose (mg)	300

## Data Availability

The datasets used during the present study are available from the corresponding author upon reasonable request.

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
