# Peer review of "Managing Severe Adverse Reactions to Biologicals in Severe Asthma"

_biomedicines, 2023, doi:10.3390/biomedicines11123108_

Round 1

Reviewer 1 Report

Comments and Suggestions for Authors

Dear authors,

paper is very interesting, however you should remove discussion from the abstract.

-------------------

Dear authors,

First of all, congratulation for this study. Treatment with biological agents is one of the most important for severe asthma. The oldest one Omalizumab as a cause of anaphylaxis was describe in few papers. The newest one, such as benralizumab and dupilumab up to these days are not so often a cause of hypersensitivity reaction.

Your protocols can be helpful for colleagues who treat severe asthmatics.

I have one significant comment: your abstract should be very abbreviated. Methods, results and conclusion are enough. Discussion should be only in the main paper.

Author Response

Response Reviewer 1

Dear Reviewer,

The authors of the article “Managing severe adverse reactions to biologicals in severe asthma” express deep gratitude to you for agreeing to review our work. Thank you very much for the professional analysis of our manuscript and for the remarks made. The authors have made the necessary changes to the text in accordance with the comments made by reviewers.

“paper is very interesting, however you should remove discussion from the abstract.”

Response: Abstract was changed, see the main text.

“I have one significant comment: your abstract should be very abbreviated. Methods, results and conclusion are enough. Discussion should be only in the main paper.”

Response: Abstract was changed, and discussion updated, see the main text.

Reviewer 2 Report

Comments and Suggestions for Authors

That is a good study illustrating the possibilty of allergic reactions for antiasthmatic biological drugs and their management. It is interesting to note that the cases reported by authors are not the lonely reported in literature. Desensitization to Omalizumab, for instance, has been reported by Petrov et al. (Current Drug Safety 2011, 6: 339-342 - Bentham Ed.) and Bernaola et al who successfully desensitized 12 patients with Omalizymab hypersensitivity (see JACI In Practice 2021, 9: 2505-8), however the authors do not mention them in the discussion. As far as Omalizumab is still concerned, polysorbates sometimes can be responsuble for the hypersensitivity reaction to the mAb, but such issue is not considered by authors (see: Ann Allergy Asthma Immunol 2018, 120: 664-6 and JInvest Allergol Clin Immunol 2020, 30: 285-287). In my opinion, these cases should be reported in the discussion with an updated authors point of view

Author Response

Response Reviewer 2

Dear Reviewer,

The authors of the article “Managing severe adverse reactions to biologicals in severe asthma” express deep gratitude to you for agreeing to review our work. Thank you very much for the professional analysis of our manuscript and for the remarks made. The authors have made the necessary changes to the text in accordance with the comments made by reviewers.

“That is a good study illustrating the possibilty of allergic reactions for antiasthmatic biological drugs and their management. It is interesting to note that the cases reported by authors are not the lonely reported in literature. Desensitization to Omalizumab, for instance, has been reported by Petrov et al. (Current Drug Safety 2011, 6: 339-342 - Bentham Ed.) and Bernaola et al who successfully desensitized 12 patients with Omalizymab hypersensitivity (see JACI In Practice 2021, 9: 2505-8), however the authors do not mention them in the discussion. As far as Omalizumab is still concerned, polysorbates sometimes can be responsuble for the hypersensitivity reaction to the mAb, but such issue is not considered by authors (see: Ann Allergy Asthma Immunol 2018, 120: 664-6 and JInvest Allergol Clin Immunol 2020, 30: 285-287). In my opinion, these cases should be reported in the discussion with an updated authors point of view”

Response:

The papers recommended were studied and added to the paper in the discussion as mentioned in the main text lines 413,437, in which we discus also the polysorbate 80 problem. References:30,31,38,39.

Thank you for this valuable observation!

Reviewer 3 Report

Comments and Suggestions for Authors

I find the manuscript to be quite interesting. However, I believe it could greatly benefit from a bit of tidying up to enhance its presentation. The current draft appears scruffy, which detracts from the insightful material within. A cleaner, more polished presentation would undoubtedly help in capturing and maintaining the reader's attention.

There is no stated aim in this article (it should be in the abstract and at the end of the introduction).

Abbreviations are not explained, eg. IgE, TNF, IFN-gamma, MoAb, FEV1,  etc.

IF authors write about some biologicals, they should briefly explain it. E.g. Omalizumab is a monoclonal antibody that specifically targets immunoglobulin E (IgE).

This manuscript should start with an introduction to which biologicals are used in asthma treatment, referenced with guidelines. E.g. https://onlinelibrary.wiley.com/doi/10.1111/all.14425

Table 2. Please move it so the headings are not alone on page 3.

Why Table 5 (line 137) is cited in the text before Table 2 (Line 139).

Author Response

Response Reviewer 3

Dear Reviewer,

The authors of the article “Managing severe adverse reactions to biologicals in severe asthma” express deep gratitude to you for agreeing to review our work. Thank you very much for the professional analysis of our manuscript and for the remarks made. The authors have made the necessary changes to the text in accordance with the comments made by reviewers.

“I find the manuscript to be quite interesting. However, I believe it could greatly benefit from a bit of tidying up to enhance its presentation. The current draft appears scruffy, which detracts from the insightful material within. A cleaner, more polished presentation would undoubtedly help in capturing and maintaining the reader's attention.”

Response: The revised version of our manuscript was carefully checked in order to harmonize the presentation.

“There is no stated aim in this article (it should be in the abstract and at the end of the introduction).”

Response: At the end of Introduction aim of the study was added, see line 121.

“Abbreviations are not explained, eg. IgE, TNF, IFN-gamma, MoAb, FEV1,  etc.”

Response: Correction made in the main text.

“IF authors write about some biologicals, they should briefly explain it. E.g. Omalizumab is a monoclonal antibody that specifically targets immunoglobulin E (IgE).”

“This manuscript should start with an introduction to which biologicals are used in asthma treatment, referenced with guidelines. E.g. https://onlinelibrary.wiley.com/doi/10.1111/all.14425”

Response: We added in introduction line 117 explanation for biologicals in severe asthma according to GINA guideline and EAACI position paper see References 9 and 10

“Table 2. Please move it so the headings are not alone on page 3.”

Response: Table 2 changed the position

“Why Table 5 (line 137) is cited in the text before Table 2 (Line 139)”

Response: We modified the Tables numbers according to the text mention.

Round 2

Reviewer 3 Report

Comments and Suggestions for Authors

All the comments have been addressed properly. Thank you.

Author Response

The authors of the article “Managing severe adverse reactions to biologicals in severe asthma” express deep gratitude to you for agreeing to review our work. Thank you very much for the professional analysis of our manuscript and for the remarks made.